# The Role of STAMP2 in Pathogenesis of Chronic Diseases Focusing on Nonalcoholic Fatty Liver Disease: A Review

**DOI:** 10.3390/biomedicines10092082

**Published:** 2022-08-26

**Authors:** Hye Young Kim, Young Hyun Yoo

**Affiliations:** Department of Anatomy and Cell Biology and BK21 Program, Department of Translational Biomedical Science, College of Medicine, Dong-A University, Busan 49201, Korea

**Keywords:** NAFLD, STAMP2, metalloreductase, iron homeostasis

## Abstract

Nonalcoholic fatty liver disease (NAFLD) is a major health issue. NAFLD can progress from simple hepatic steatosis to nonalcoholic steatohepatitis (NASH). NASH can progress to cirrhosis or hepatocellular carcinoma. Unfortunately, there is no currently approved pharmacologic therapy for NAFLD patients. The six transmembrane protein of prostate 2 (STAMP2), a metalloreductase involved in iron and copper homeostasis, is well known for its critical role in the coordination of glucose/lipid metabolism and inflammation in metabolic tissues. We previously demonstrated that hepatic STAMP2 could be a suitable therapeutic target for NAFLD. In this review, we discuss the emerging role of STAMP2 in the dysregulation of iron metabolism events leading to NAFLD and suggest therapeutic strategies targeting STAMP2.

## 1. Introduction

Nonalcoholic fatty liver disease (NAFLD) is the main cause of liver disease worldwide. NAFLD is speculated to become the leading cause of end-stage liver disease in the future. NAFLD affects both adults and children [1]. Nonalcoholic steatohepatitis (NASH) is an aggressive form of NAFLD. NASH can progress to cirrhosis and hepatocellular cancer (HCC). NASH is rapidly becoming the leading cause of end-stage liver disease or liver transplantation [2,3]. Currently, no pharmacotherapies are approved for the treatment of NAFLD. The current treatment is lifestyle modification, which is based on bodyweight and exercise. Various novel therapeutic approaches are under clinical development. Despite decades of intensive study on the pathogenesis of NAFLD, understanding the mechanism underlying the pathogenesis of this disease is still incomplete.

Six transmembrane protein of prostate 2 (STAMP2) exerts a pivotal function in metabolic and inflammatory pathways. Previous studies elucidated that altered STAMP2 expression disrupts metabolic homeostasis [4,5,6,7]. Additionally, various studies have shown that dysregulation of STAMP2 is associated with metabolic and inflammatory diseases, such as obesity [8,9,10], rheumatoid arthritis [11,12,13,14], and atherosclerosis [15,16]. In addition, recent epidemiological research has reported an association between single nucleotide polymorphisms (SNPs) in the STAMP2 gene and metabolic syndrome [17,18,19]. Emerging evidence has further demonstrated that knockdown or elimination of STAMP2 results in various metabolic and inflammatory dysregulations. Adipose tissue of STAMP2-/- mice exhibited the expression of inflammatory genes as well as impaired insulin action [10]. STAMP2 knockdown resulted in insulin-stimulated glucose uptake and GLUT4 translocation [20]. Elimination of STAMP2 using monoclonal antibody ameliorated high glucose and S100B-induced effects in diabetic mouse kidney [21]. In contrast, the STAMP2 overexpressing vector reduced inflammation and ameliorated glucose metabolism in a mouse model of streptozotocin-induced diabetes [22]. Another study undertaken by adenoviral delivery of STAMP2 to diabetic ApoE-/-/LDLR-/- mice showed similar phenotypes. Adenoviral STAMP2 ameliorated insulin resistance through the reduced macrophage infiltration and the reversal of the increased expression of proinflammatory cytokines in epididymal and brown adipose tissue (BAT) [23].

We previously showed that hepatic STAMP2 plays a pivotal role in alleviating either high-fat-diet (HFD)- or environmental-pollutant (polychlorinated biphenyl, PCB)-induced NAFLD and upregulation of STAMP2 using adenoviral STAMP2 ameliorates hepatic steatosis, inflammation, and iron overload in NAFLD, in in vivo and in vitro models [24,25,26,27,28]. Structurally, STAMP2 has metalloreductase activity and functions in iron and copper homeostasis [29,30]. However, how the metalloreductase action of STAMP2 is related to the metabolic protective effect of STAMP2 is still unknown. In this review, we discuss the emerging role of STAMP2 in the dysregulation of iron metabolism events leading to NAFLD and suggest therapeutic strategies targeting STAMP2.

## 2. Nomenclature of STAMP2

Through multiple different approaches, the STAMP (six transmembrane protein of prostate) protein family has been studied [31,32,33]. Initially, it was considered a prostate cancer marker. Thus, the protein family was named STEAP (six transmembrane epithelial antigen of the prostate) [32]. Other studies of prostate-specific antigens designated the protein as STAMP [33,34]. Now, both STAMP and STEAP are being used: STAMP1, STAMP2, and STAMP3 correspond to STEAP2, STEAP4, and STEAP3, respectively. A previous study isolated a tumor necrosis factor α (TNF-α)-inducible protein in rat (*Rattus norvegicus*) in adipose tissues and confirmed the involvement of a six transmembrane cell surface protein, which was named TIARP (tumor necrosis factor α-induced adipose-related protein) [31]. Subsequently, because of high homology between TIARP and its STEAP counterparts in humans (*Homo sapiens*), it was renamed STEAP4. Although STAMP2, STEAP4, and TIARP are being used interchangeably, TIARP, STEAP4, and STAMP2 are not completely identical in function. For example, whereas TIARP regulates preadipocyte differentiation in vitro, STEAP4 does not [5]. Accordingly, further future study of these species-specific differences is a challenging task.

## 3. Structure and Function of STAMP2

Molecular cloning revealed that the STAMP2 gene is located on chromosome 7q21 and contains five exons and four introns and that the genomic sequence is relatively small (approximately 26 kb) because intron 1 is extremely large (22,516 bp). The study also showed that the STAMP2 gene is transcribed into a single 4.0 kb mRNA with a 5′-untranslated region (UTR) of approximately 1.7 kb [33]. The protein comprises 495 amino acids and consists of an N-terminal cytoplasmic oxidoreductase domain (OxRD) and a six-helical transmembrane domain (TMD) near the C-terminal domain. Furthermore, a conserved domain search identified three motifs at the N-terminal domain: a dinucleotide binding domain, an NADP oxidoreductase motif, and a motif similar to pyrroline 5-carboxylate reductase [33].

### 3.1. Cellular Location

It was shown by quantitative time-lapse and immunofluorescence confocal microscopy using STAMP2 labeled with an N-terminal green fluorescent protein (GFP) tag that STAMP2 was detected predominantly in the plasma membrane and also cytoplasmic vesicles associated with the Golgi apparatus. Therefore, STAMP2 appears to localize to plasma membrane, the Golgi complex, and the trans-Golgi network [31,33]. Immunoblot analysis using intracellular fractions showed that STAMP2 was localized in early endosomes and mitochondria [11,35]. Recently, exon 3-spliced variant STEAP4 (v-STEAP4) has been found to be highly expressed in porcine lung and in HepG2 cells derived from human liver carcinoma cells [36]. Another group identified v-STEAP4 in CD14^+^ monocytes from patients with rheumatoid arthritis (RA) and suggested that v-STEAP4 was expressed in the nuclear fraction of THP-1 cells overexpressing v-STEAP4 [12,37].

### 3.2. Crystal Structures and Functional Motifs

The crystallographic and kinetic characterization of the murine Stamp2 oxidoreductase domain demonstrated that STAMP2 utilizes an interdomain flavin-binding site to shuttle electrons between the NADPH-utilizing oxidoreductase domain and the transmembrane heme group [38]. The single-particle cryo-electron microscopy on human STAMP2 demonstrated the trimeric structures reveal an aligned, inter-subunit NADPH-FAD-heme arrangement, suggesting that chelated iron (III) binds in a cavity of basic residues to facilitate reduction [39].

Even in the presence of various mutations, it is noticeable the STAMP2 *K_m_* value is stable. Presumably, the three-dimensional structure, not a single conserved residue, is important for the protein’s activity. This is consistent with a previous epidemiological study of insulin resistance syndrome (DESIR), which showed that common polymorphisms (SNPs) of STAMP2 had little correlation with metabolic syndrome [18].

### 3.3. Metalloreductase Activity

The STAMP2/STEAP4 protein is an integral membrane metalloreductase. STAMP2 moves electrons from intracellular NADPH to extracellular metal iron or copper through FAD and a single heme [29,40]. For the transport of extracellular Fe^3+^ and Cu^2+^ across the membrane into the cell, reductions in these metals are required. The STEAP family reduces Fe^3+^ and Cu^2+^ ions to facilitate metal-ion uptake by mammalian cells. Interestingly, STEAPs are the only delineated eukaryotic cytochromes known to perform transmembrane electron transport using a single-heme ligand.

STAMP2 shows the highest iron uptake values of any member of the STEAP family [40]. STAMP2 exhibits ferric-reductase activity in the presence of its redox cofactors FAD and heme and the electron donor NADPH. Based on these reports, the enzyme does not require specific accessory proteins for catalysis [39]. Having employed cryo-electron microscopy, the study revealed the molecular mechanism of iron reduction and the steps through which STAMP2 reduces metal iron. This strongly suggests that STAMP2 may be critical to both iron and copper homeostasis at the cellular level and within the body [39].

Metalloreductase is involved in a variety of biological processes, such as modulating oxidative stress, reducing metals for cellular uptake, and functioning as a coenzyme (in this case, for NADPH) in metabolic regulation. Previous studies indicate that STAMP2 performs all of these functions [16,30,41]. Furthermore, STAMP2 is associated with metabolic diseases [7,10,25,42] and is overexpressed in several human cancers [32,43,44,45], underlining its physiological function in the maintenance of cellular iron homeostasis [30,42].

### 3.4. Endocytic and Exocytic Pathway

Live cell imaging of GFP-tagged STAMP2 shows that it is present in vesiculo-tubular structures (VTSs). VTSs are structures to shuttle between the plasma membrane and the trans-Golgi network (TGN)/Golgi/ER. Some of the VTSs move in curvilinear paths and others in straight ones [33]. STAMP2 also colocalizes with caveolin-1 in 3T3-L1 adipocytes [46]. Furthermore, GFP-STAMP2 colocalizes significantly with an early endosome-associated protein EEA1 [47]. STAMP2 could be active in intracellular organelles based on that it exhibits a fairly stable *K_m_* value under acidic conditions [38]. Although the exact role of STAMP2 in molecular trafficking is unknown, it may be involved in endocytic and/or exocytic pathways. STAMP2, in an unfolded state, may reach the endoplasmic reticulum and then acquire its active conformation.

## 4. Regulation of STAMP2 Expression

STAMP2 expression has been found in multiple tissues and is regulated by a number of different stimuli. The STAMP2 gene is mainly expressed in adipose tissue, placenta, bone marrow, lung, pancreas, and heart, followed by prostate, liver, skeletal muscle, pancreas, testis, small intestine, and thymus, with no detectable expression in the central nervous system [31,33]. Analysis of metabolic tissues revealed that STAMP2 is found in adipose tissue, hepatocytes, and pancreatic islets/beta-cells [7,33]. Several studies have suggested that STAMP2 is highly expressed joints of patients with rheumatoid arthritis (RA) [33,48,49]. Accumulating evidence has indicated that STAMP2 is regulated by multiple factors, including cytokines, nutrients, hormones, and transcription factors (Table 1). In our previous study, we performed high-throughput drug screening using small-molecule libraries consisting of 2000 diverse compounds to identify a STAMP2 enhancer and found 18 compounds as STAMP2 modulators: 13 upregulators and 5 downregulators [27]. Furthermore, we also showed that cilostazol and recombinant FGF21 enhance the expression of hepatic STAMP2 [26,27].

## 5. STAMP2 in Type 2 Diabetes, Inflammatory Diseases, and Cancers

STAMP2 is a critical modulator for coordinating metabolism and inflammation [6,7,10]. STAMP2 is significantly downregulated in the adipose tissue of obese patients [4] and in *ob/ob* mice or HFD-induced obese mice [10] and is involved in tissue metabolic regulation, such as improving glucose uptake, decreasing the inflammatory response, and increasing insulin sensitivity. Although STAMP2 has been widely studied focusing on the inhibitory role in inflammation and metabolism, the underlying mechanism is not fully understood. In addition to its role in metabolism and inflammation, STAMP2 is also associated with tumorigenesis. For example, STAMP2 overexpression may increase ROS, which may contribute to increased mutational rates and further progression of prostate cancer [33,43,57].

Therefore, while STAMP2 may have a beneficial role associated with chronic metabolic and inflammatory diseases [4,6,10], dysregulation of STAMP2 expression may also promote cancer cell proliferation and cancer progression [58,59]. These reports suggest various roles for STAMP2 in health and disease.

### 5.1. STAMP2 and Type 2 Diabetes

NAFLD is present in >70% of individuals with type 2 diabetes [60]. Impairment of glucose and lipid metabolism, which has been accelerated by the worldwide increase in the prevalence of obesity and type 2 diabetes, is most likely behind the increase in patients with NAFLD [61]. Insulin resistance underlies both obesity and type 2 diabetes. Insulin activates the PI3 K/Akt pathway, which is responsible for insulin-stimulated glucose uptake. In response to insulin, phosphatidylinositol 3-kinase (PI3 K) is activated, which leads to the phosphorylation of Akt. Phosphorylated Akt induces GLUT4 translocation to the plasma membrane, which directly increases glucose transport into cells [62]. Previous studies showed that STAMP2 affects insulin-stimulated GLUT4 translocation and glucose transport by targeting the PI3 K/Akt signaling pathway in human adipocytes [20,21]. Cheng et al. found that STAMP2 deficiency significantly reduced GLUT4 translocation, glucose uptake, and the phosphorylation levels of PI3 K (P85) and Akt [20]. A loss of function study using a STAMP2 antibody also showed a decrease in the insulin-stimulated tyrosine phosphorylation of the insulin receptor substrate (IRS)-1, phosphorylation of PI3 K (P85), and Akt [21]. JNK is an important mediator of insulin resistance too. Activation of JNK decreases insulin activity [62]. In diabetic ApoE-/-/LDLR-/- mice, the phospho-JNK/JNK ratio was increased in white adipose tissue (WAT) and BAT, but overexpression of STAMP2 significantly decreased the ratio of phospho-JNK/JNK [23]. In our previous study, while liver-specific knockdown of Stamp2 by in vivo siRNA delivery increased insulin resistance, overexpression of hepatic STAMP2 improved HFD-induced insulin resistance [25]. These results suggest that the STAMP2 gene is involved in the pathogenesis of type 2 diabetes.

### 5.2. STAMP2 and Inflammatory Diseases

Dysregulation of STAMP2 has been implicated in various inflammatory diseases, including obesity [8,9,10,18], rheumatoid arthritis (RA) [11,14,48], and atherosclerosis [15,16]. The expression of TNFα in the synovia correlates with the progression of joint swelling in both murine models and arthritis patients. STAMP2 expression is observed in the synovium of rheumatoid arthritis. STAMP2 is induced in synovia by TNFα [11,14,48]. In addition, RA-like pathology was observed in Stamp2 knockout mice [11]. A marked increase in atherosclerotic lesion area was demonstrated in the aortas of Stamp2-/-ApoE-/- mice, a model system to study atherosclerosis. Stamp2 is detected in mouse and human atherosclerotic plaques and its deficiency promotes atherosclerosis in mice. These findings suggest a role for STAMP2 in protecting against atherosclerosis [16]. STAMP2 expression was previously detected in circulating monocytes and its expression correlated with the macrophage marker CD68 [8]. Furthermore, adenoviral overexpression of Stamp2 in ApoE-/-LDLR-/- diabetic mice suppressed atherosclerosis by preventing macrophage apoptosis [15].

STAMP2 integrates inflammatory and nutritional signals with metabolism, which is supported by the loss of STAMP2 function, both in vitro and in vivo results, in elevated inflammatory markers, diminished insulin sensitivity, and dysfunctional glucose uptake. Stamp2 knockout mice also exhibit an increased number of macrophages [6,10,63]. STAMP2 may counter-regulate insulin resistance through regulating macrophage polarization in visceral adipose tissue (VAT) and BAT [23]. In contrast, STAMP2 overexpression actively protects adipocytes against inflammatory challenges. STAMP2 overexpression reduces rates of atherosclerosis and plaque formation in diabetic mice, while STAMP2 deficiency promotes atherosclerosis [15]. Similarly, STAMP2 overexpression reduces the migration of neutrophil-like HL60 cells [64] and reduces IL-6 and IL-8 cytokine expression, whereas siRNA knockdown of STAMP2 increases cytokine signaling in patients with rheumatoid arthritis (RA) [11], again, consistent with a role for STAMP2 in a negative feedback loop. These protective effects have also been observed at the systemic level. STAMP2 also exerts anti-inflammatory activity in other tissues, most notably, the liver. Hepatic STAMP2 appears to be a target of STAT3, which is known for negatively regulating hepatic gluconeogenic gene expression, thus, playing a protective role in hepatic insulin signaling [52,65]. This suggests that STAMP2 conducts protective activity in maintaining insulin signaling in the presence of obesity and inflammation signals. STAMP2 also controls macrophage inflammation by controlling NADPH homeostasis [16].

### 5.3. STAMP2 and Cancers

STAMP2 was originally identified as a gene that had sequence similarity to STAMP1 and high expression in the prostate. It was then found to be regulated by androgens, with increased mRNA levels in PCa cells compared with normal prostate cells [33]. Several studies revealed the association of STAMP2 with prostate cancer. STAMP2 overexpression induced an increase in number and size of the PC-3 cell line and increased growth of COS7 and DU145 cells [33]. STAMP2 knockdown induced apoptosis and cell cycle arrest and markedly inhibited the growth of PCa cell lines, including LNCaP, VCaP, and 22 Rv1, both in vitro and in vivo [43]. Additionally, another recent study suggested that STAMP2 knockdown inhibits the proliferation of prostate cancer cells through the activation of the cGMP-PKG pathway in an inflammatory microenvironment [66].

STAMP2 has been recently reported to be involved in other cancers as well. Wu et al. found in keratinocytes that STAMP2 expression is regulated by IL-17 to be known to promote cancer [53]. STAMP2 was critical for IL-17-induced proliferation in cultured keratinocytes. STAMP2 was required for IL-17-dependent sustained activation of the TRAF4-ERK5 axis for keratinocyte proliferation and tumor formation. Further work is needed to understand the details of how STAMP2 affects keratinocyte biology and tumorigenesis. Orfanou et al. found that STEAP4 expression was obvious only in malignant breast tissues, whereas all benign breast cases had no detectable levels. Furthermore, knockdown of STEAP4 also suppressed cell proliferation and enhanced the pharmacological effect of lapatinib (HER2 inhibitor) in HER2-overexpressing breast cancer [45], confirming its potential oncogenic role in breast cancer. These results in various cancers demonstrate that the growth-inducing and anti-apoptotic roles of STAMP2 are needed for carcinogenesis. On the other hand, another study investigated RNA-seq in hepatocellular carcinoma (HCC) between GSE54503 and TCGA datasets and showed that STAMP2 expression was reduced in HCC tissues and that STAMP2 inhibited the proliferation and metastasis of HCC cells. These findings indicate that STAMP2 functions as a tumor suppressor gene in HCC [67].

Therefore, the differential expression of STAMP2 in normal and cancer tissue makes STAMP2 a potential candidate as a biomarker or a therapeutic target for cancer.

## 6. Pathogenesis of NAFLD and Therapeutic Approaches 

The pathogenesis of NAFLD is a multifactorial and complex process. The altered intrahepatic regulation of free fatty acid uptake, synthesis, degradation, and secretion leads to the accumulation of triglycerides in hepatocytes. Changes cause the liver to be susceptible to injury from inflammatory responses, which aid in the progression of the disease. Additionally, there is a strong relationship between NAFLD and the components of metabolic syndrome, including type 2 diabetes mellitus (T2DM). Up to 70% of patients with T2DM may have concurrent NAFLD. These patients are at higher risk of progression to NASH and advanced fibrosis [3,68,69,70].

### 6.1. Prevalence of NAFLD

The estimated prevalence of NAFLD is approximately 25% of the world population. The global prevalence of NAFLD is highly reported in the Middle East (32%) and South America (31%), followed by Asia (27%), the USA (24%), and Europe (23%); however, it is less common in Africa (14%) [71]. The estimated prevalence of steatosis is 45% in the Hispanic population, 33% in the Caucasian population, and 24% in the African-American population. In terms of gender differences, the prevalence is approximately 42% for men and 24% for women [72]. Remarkably, children are recognized to be at risk of developing NAFLD as well: the prevalence is almost 3–10% in lean children and approximately 53% in obese pediatric populations [73]. In addition, from a pathological point of view, given the increasing global epidemic of obesity and T2DM, a recent model has estimated a 178% increase in liver deaths related to NASH by 2030 [71].

### 6.2. Pathogenesis of NAFLD

Recently, international consensus has proposed the term metabolic-associated fatty liver disease (MAFLD) [74], as the presence of hepatic steatosis plus one of the following: overweight/obese, T2DM, or evidence of two or more features of metabolic dysfunction. Day and James first proposed the “two-hit” hypothesis in 1998 [75]. Essentially, the ‘‘first hit’’ occurs when hepatic steatosis is induced via lipid accumulation in the hepatocytes due to IR, which increases the liver vulnerability to “second hits” that subsequently initiate the development of the inflammatory, fibrosis, and cellular death characteristics of NASH. The second hit can be a variety of factors, such as oxidative stress, proinflammatory cytokines, endoplasmic reticulum (ER) stress, and gut-derived bacterial endotoxin [2,3,69]. The initial “two-hit” theory can no longer completely explain the pathogenesis of NAFLD, which involves multiple factors. Thus, the historical ‘‘two-hit’’ pathogenic theory has been switched to a multifactorial model, the “multiple parallel hits theory”, which more precisely summarizes the complex features of NAFLD pathogenesis [76,77,78]. 

The global epidemic of NAFLD has largely been attributed to endogenous factors, such as diet, genetics, lifestyles, and aging. However, there is also increasing evidence of the contributing role of environmental exposure [79,80]. The coined term toxicant-associated steatohepatitis (TASH) has emerged to reflect the fact that a form of NAFLD/NASH observed in workers with high exposure to toxic pollutants is not rare [81].

### 6.3. Current Therapeutic Approaches to NAFLD

Therapeutic approaches for NAFLD are difficult due to the complex etiology, problematic diagnosis, divergent spectrum of its stages, and possibility for coexisting diseases. As our understanding has progressed through recent studies focusing on epidemiology, diagnosis, and treatments, novel biomarkers and treatment strategies have begun to emerge.

Both genetic and lifestyle factors appear to contribute to the pathogenesis of NAFLD. Further, at least at the early stages of the disease, lifestyle changes—namely, improved diet, increased physical activity, and weight management—can be an effective strategy to manage NAFLD [82]. In many cases, however, pharmacological intervention may be necessary. A number of the reported therapeutic strategies have designated insulin sensitizers, lipid-lowering agents, antioxidants, and cytoprotective agents as therapeutic measures against multifaceted NAFLD pathways. However, to date, no pharmacological therapy has been approved for the treatment of NAFLD. All the proposed potential pharmacotherapies are prescribed for NAFLD management, including treating the associated obesity, oxidative stress, T2DM, hyperlipidemia, and inflammation (Table 2). Therefore, advances in targeted drug therapy are a priority.

## 7. Role of STAMP2 in NAFLD Pathogenesis and Therapeutic Strategies

The pathogenesis of NAFLD/NASH is considered to be a two-hit result of hepatic steatosis and inflammation [77,83]. In adipocytes, STAMP2 has been identified as a counterregulatory protein of inflammation and insulin resistance [4,6,7]. STAMP2-/- mice display elevated expression of proinflammatory mediators in WAT and impairment of insulin-stimulated glucose transport in adipocytes [10]. Knockdown of STAMP2 inhibits insulin-stimulated glucose transport and GLUT4 translocation through attenuated phosphorylation of Akt [20]. In hepatocytes, Wellen et al. found that the lack of STAMP2 correlates with dysfunctional responses to fat and nutrient influx and the onset of NAFLD [10]. On the other hand, the expression of STAMP2 in hepatocytes has been suggested to suppress lipogenesis and gluconeogenesis [36]. In our previous study, we found that the expression level of STAMP2 protein was significantly reduced in the livers obtained from NAFLD patients and HFD-induced NAFLD mice. In these mice, liver-specific deletion of Stamp2 by in vivo siRNA delivery accelerated hepatic steatosis, as indicated by markedly increased vacuolization, increased liver weight, elevated plasma total cholesterol, triglyceride and nonesterified fatty acid levels, and increased insulin resistance. Conversely, overexpression of hepatic Stamp2 using adenoviral delivery improved these effects. Hepatic STAMP2 alleviated HFD-induced steatosis through downregulation of lipogenic and adipogenic transcription factors, sterol response element binding protein 1 (SREBP1), and peroxisome proliferator-activated receptor (PPARγ) [25]. Overall, it was thought that STAMP2 prevents NAFLD development by repressing lipogenic and adipogenic factors and modulating insulin signaling. Additionally, hepatic STAMP2 overexpression decreases hepatitis B virus X-protein signaling and subsequent metabolic dysfunction [24]. Thus, STAMP2 appears to play a protective role against metabolic and inflammatory stresses. We previously demonstrated that hepatic STAMP2 plays a pivotal role in preventing HFD-induced NAFLD and that both cilostazol and recombinant fibroblast growth factor 21 (FGF21) ameliorate HFD-induced hepatic steatosis by enhancing hepatic STAMP2 expression through AMPK [25,26,27]. These studies demonstrate that STAMP2 could be a suitable target for NAFLD patients.

## 8. STAMP2 Is a Hepatic-Iron-Overload-Targeted Modulator of NAFLD

### 8.1. Iron Homeostasis and NAFLD

Several metabolic components involved in glucose and lipid metabolism require iron as a cofactor [84,85]. The dysregulation of iron may cause insulin resistance and lipid imbalances in the liver. Recent studies suggest that dysregulation of iron metabolism is involved in the development of insulin resistance, dyslipidemia, and NAFLD/NASH [84,85,86]. Furthermore, various studies have reported an association of iron with more advanced stages or a higher incidence of NAFLD/NASH [87,88,89]. The liver is one of the most critical organs for iron storage. Approximately 25–30% of total iron in the body is stored in ferritin in the liver and the intrahepatic iron content [90].

In terms of the disruption of iron homeostasis in NAFLD, there have been some conflicting reports [91,92,93]. Dysmetabolic iron overload syndrome (DIOS) can be found in up to one-third of patients with NAFLD and is characterized by metabolic syndrome and high serum ferritin (SF) and transferrin-iron saturation (TS) levels, but normal serum iron levels [94]. In NAFLD/NASH animal models, mice that develop hepatic iron overload (HIO) after being fed dietary iron can exhibit hepatic oxidative stress, inflammasome activation, upregulation of inflammatory and immune mediators, and hepatocellular ballooning [95]. In humans, the pattern of HIO has been associated with NAFLD severity [96]. Moreover, some reports have demonstrated that iron reduction therapies, such as phlebotomy and iron chelation, improve insulin resistance and liver function in patients with NAFLD [97,98]. However, other reports have failed to find an increase in hepatic iron deposition in NAFLD and studies of phlebotomy in patients with NAFLD have been generally negative [99,100,101,102].

### 8.2. STAMP2 and Iron Homeostasis

In mammals, iron mainly exists in its oxidized form (Fe^3+^) bound to the carrier protein transferrin in serum. However, iron uptake by cells requires ferrous iron (Fe^2+^) because metal transporters specifically import divalent cations [84]. STAMP2, as a ferrireductase, can reduce Fe^3+^ to Fe^2+^, which allows its transport out of the endosome by divalent metal transporter 1 (DMT1) [40]. Overexpression of mouse STAMP2 in HEK-293 cells stimulates cellular uptake of ferrous iron. According to this assay, STAMP2 shows the highest iron uptake values of any member of the STEAP family [40,103]. This strongly indicates that STAMP2 plays a role in the cellular uptake of iron. Accordingly, STAMP2 seems to be critical to both iron homeostasis at the cellular level and within the body. Several other regulatory genes in iron metabolism, such as lipocalin-2 [104], hepcidin, and ferritin [105], have been recognized as important to both inflammation and metabolic disorders. However, how STAMP2 itself or in cooperation with these factors regulates metabolic homeostasis has not yet been elucidated.

### 8.3. STAMP2 as a Potential Therapeutic Target for NAFLD Accompanying HIO

STAMP2 is an iron-metabolism-related protein that functions as a metalloreductase involved in cellular iron and copper homeostasis and its expression is modulated in response to inflammatory stimuli. NAFLD may proceed to NASH characterized by steatosis, inflammation, and oxidative stress [83]. Our previous study demonstrated that hepatic STAMP2 mediates recombinant FGF21-induced improvements in hepatic iron overload in nonalcoholic fatty liver disease through the upregulation of expression of the iron exporter, ferroportin [26]. In addition, we have also shown that hepatic STAMP2 alleviates polychlorinated biphenyl (PCB)-induced steatosis in NAFLD in in vivo and in vitro models [28]. These studies support that STAMP2 targeting could be a potential strategy for NAFLD accompanying HIO.

## 9. Concluding Remarks and Future Perspectives

An important aspect in recent NAFLD research is the elucidation of links between glucose/lipid metabolism and inflammatory signaling pathways. STAMP2 has emerged as a key player in glucose/lipid metabolism and inflammatory responses in metabolic tissues. The STAMP2 protein has metalloreductase activity, which may be important for STAMP2 function. Despite this apparent link, the function of STAMP2 as a metalloreductase has not been extensively investigated in the context of metabolic diseases. A better understanding of STAMP2 and its interaction with other important pathways in all types of cells involved in the NAFLD spectrum may lead to the development of new diagnostic or prognostic biomarkers or therapeutic targets for NAFLD.

## Figures and Tables

**Table 1 biomedicines-10-02082-t001:** Stimuli/factors that can regulate expression of STAMP2.

Stimuli/Factors		Effect	Tissues/Cell Lines	References
Cytokines	TNFα	↑	3T3-L1 murine adipocytes, mouse adipose tissue, human adipocytes, MH7A human synovial cell	[5,10,11,31,50,51]
	IL-6	↑	3T3-L1 murine adipocytes, murine adipose and liver tissue, human adipocytes	[5,49,51]
	IL-1β	↑	3T3-L1 murine adipocytes, differentiated human mesenchymal stem cells	[52]
	IL-17	↑	Murine keratinocyte	[53]
	Leptin	↓	Human adipocytes	[5]
Nutritional status	Obese	↑	Human: Visceral and subcutaneous adipose tissueMurine: Liver of HFD or genetically induced miceRabbit: Prostate in animals on high fat diet	[8,49,54,55]
		↓	Human: Visceral adipose tissue, livers obtained from NAFLD patients Murine: Liver, visceral and brown adipose tissue of HFD or genetically induced mice	[9,23,25,26,27,49,50]
	Feeding	↑	Murine: Liver, and visceral, subcutaneous and brown adipose tissue	[10,49]
	Fasting	↓	Murine: Liver, Skeletal Muscle, and visceral, subcutaneous and brown adipose tissue	[10,49]
Nutrients	Glucose	↑	MES13 murine mesangial cells	[22]
	Oleic acid	↑	3T3-L1 murine adipocytes	[10]
		↓	Liver tissue of mice on HFD, primary hepatocyte, HepG2 cells	[25,26,27]
	High serum	↑	3T3-L1 murine adipocytes	[10]
Hormones	Androgen	↑	Human prostate cancer cells: LNCaP, VCaP, 22Rv1 Visceral adipose tissue of rabbits on HFD	[33,43,56]
		↓	Prostate tissue of rabbits fed on high fat diet	[55]
	insulin	↓	3T3-L1 murine adipocytes	[51]
	Growth hormone	↑	3T3-L1 murine adipocytes	[51]
Transcription factors	C/EBPα	↑	Murine: Adipocytes, liver, HepG2 cells	[10,49]
	C/EBPβ	↑	HepG2 cells	[36]
	STAT3	↑	MES13 cells	[22,49]
	LXRα	↑	3T3-L1 murine adipocytes	[10]
Kinases	AMPK	↑	Murine: Liver, HepG2 cells	[26,27]
	JNK	↑	MES13 cells	[22]
	PI3K	↑	MES13 cells	[22]
	JAK2	↑	MES13 cells	[22]
Etc.	Environmental disruptors, PCBs	↓	Murine: Liver, primary hepatocytes, HepG2 cells	[28]
	LPS	↑	Primary peritoneal macrophages, HepG2 cells	[16,36]

**Table 2 biomedicines-10-02082-t002:** Current pharmacotherapeutic strategies in NAFLD treatment.

Therapeutic Agents		MOA	Therapeutic Agents		MOA
Antioxidants	Vitamin E	antioxidant	Anti-hyperlipidemic agents	Statins	hypolipidemic action
	UDCA	cytoprotective agent		Fibrates	hypolipidemic action
	SAMe	antioxidant		Fenofibrate	hypolipidemic action
	Betaine	antioxidant		Gemfibrozil	hypolipidemic action
	N-acetylcysteine (NAC)	protects against oxidative stress		Ezetimibe	hypolipidemic action
Insulin sensitizers	Metformin	biguanide	Anti-inflammatory drugs	OCA, Cilofexor, EDP-305, EYP 001	FXR agonists
	Thiazolidinediones (TZD)	PPARγ agonist		Cenicriviroc	CCR2/5 antagonist
	Pioglitazone	PPARγ agonist	Metabolic enzyme inhibitors	Aramchol	SCD1 inhibitors
	Elafibranor	dual PPARα/δ agonist		Firsocostat, PF-05221304	ACC 1/2 inhibitors
	Saroglitazar	dual PPARα/δ agonist		PF-06835919	Ketohexokinase inhibitors
	lanifibranor	pan-PPAR agonist		cilostazol	phosphodiesterase, PDE3 inhibitor
Anti-hyperglycemic agents	Liraglutide, Semaglutide	GLP-1 receptor agonist (GLP-1 RA)	Modulators of energy metabolism	VK2809, resmetirom (MGL-3196)	THRβ agonists
	Tirzepatide	Dual GLP-1/GIP RA		MSDC-0602 K	Mitochondrial pyruvate carrier inhibitors
	Sitagliptin, Vildagliptin	DPP-4 inhibitors		NGM-282	FGF19 analogs
	Empagliflozin, Canagliflozin, Dapagliflozin	SGLT2 inhibitors		Pegbelfermin (BMS-986036)	FGF21 analogs

MOA, Mechanism of action; UDCA, Ursodeoxycholic acid; SAMe, S-adenosyl-L-methionine; OCA, Obeticholic acid; GLP-1, Glucagon-like peptide-1; DPP-4, Dipeptidyl peptidase-4; SGLT-2, Sodium glucose cotransporter-2; THR, Thyroid hormone receptor; FGF, Fibroblast growth factor.

## Data Availability

Not applicable.

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
