# Peer review of "The Role of STAMP2 in Pathogenesis of Chronic Diseases Focusing on Nonalcoholic Fatty Liver Disease: A Review"

_biomedicines, 2022, doi:10.3390/biomedicines10092082_

Round 1

Reviewer 1 Report

The content of the work goes far beyond the knowledge of NAFLD and the role of STAMP2 in the regulation of iron metabolism. For this reason , please consider changing the work title, for example: The role of STAMP2 in pathogenesis of chronic diseases with focusing on Nonalcoholic Fatty Liver Disease: a Review.

Plese remove any editorial errors carefully, and complete the references (No 13,28,58,63,64,65,79,85 and other)

Author Response

Comments by reviewer #1

  1. The content of the work goes far beyond the knowledge of NAFLD and the role of STAMP2 in the regulation of iron metabolism. For this reason, please consider changing the work title, for example: The role of STAMP2 in pathogenesis of chronic diseases with focusing on Nonalcoholic Fatty Liver Disease: a Review.

(Answer) Having appreciated this critique, we changed the work title to “The Role of STAMP2 in Pathogenesis of Chronic Diseases with Focusing on Nonalcoholic Fatty Liver Disease: A Review”.

  1. Please remove any editorial errors carefully, and complete the references (No 13,28,58,63,64,65,79,85 and other)

(Answer) Having appreciated this critique, we carefully removed editorial errors and complete the references (No 13,28,58,63,64,65,79,85 and others including No 1,9,50,60,83,97,109). The revised portions are marked with “Track Changes” function in manuscript.

Reviewer 2 Report

The review article titled as "Molecular Role of STAMP2 in Nonalcoholic Fatty Liver Disease 2 Accompanying the Dysregulation of Iron Metabolism" is well compiled information and is intresting to read. There are a few changes I would suggest.

1. There are several spellings and gramatical errors and thus authors are suggested to carefuly proof read the article and make the changes.

2. In table 1, generally obesity does not comes under nutrition category, instead authors can devide this section into two, (A) nutritional status (feeding fasted, obese) and (B) Nutrients fat, glucose etc

Author Response

Comments by reviewer #2

The review article titled as "Molecular Role of STAMP2 in Nonalcoholic Fatty Liver Disease 2 Accompanying the Dysregulation of Iron Metabolism" is well compiled information and is interesting to read. There are a few changes I would suggest.

  1. There are several spellings and grammatical errors and thus authors are suggested to carefully proof read the article and make the changes.

(Answer) Having appreciated this critique, we carefully proof read the article and corrected several spellings and grammatical errors. The revised portions are marked with “Track Changes” function in manuscript.

  1. In table 1, generally obesity does not comes under nutrition category, instead authors can devide this section into two, (A) nutritional status (feeding fasted, obese) and (B) Nutrients fat, glucose etc.

(Answer) Having appreciated this critique, we modified Table 1 in manuscript. We divided mentioned section into two, Nutritional status (obese, feeding, fasting) and Nutrients (glucose, oleic acid, high serum) as suggested by this reviewer.